# Local Administration of ElectroMagnetic Field as Add-On Therapy in the Treatment of Chronic Facial Pain: A Pilot Study

**DOI:** 10.3390/ijerph19074123

**Published:** 2022-03-30

**Authors:** Marco Storari, Nicoletta Zerman, Enrico Spinas

**Affiliations:** 1Department of Surgical Science, College of Dentistry, University of Cagliari, 09124 Cagliari, Italy; m.storari93@gmail.com; 2Department of Surgery, Dentistry, Pediatrics and Gynaecology, University of Verona, 37124 Verona, Italy; nicoletta.zerman@univr.it

**Keywords:** chronic pain, electromagnetic field, fibromyalgia, orofacial neuropathic pain

## Abstract

Fibromyalgic syndrome and orofacial neuropathic pain are major public health concerns affecting up to 5% and 10%, respectively, of the general population. They generally require medications such as antidepressants and anticonvulsants, which may additionally impact the quality of life with their side effects. Modern technologies and related applications have changed several fields of human life, even in medicine. In the current study, the local administration of electromagnetic fields as add-on therapy for the treatment of cervical and facial pain in patients with fibromyalgia or neuropathic pain has been evaluated. A total of 15 patients were recruited, and an electromagnetic field was delivered through a small patch applied between C3 and C4. Patients were followed for 12 months, and pain levels were rated via the VAS scale; ∆% was calculated through the analysis of median VAS scale values at each time point. Mild-to-moderate improvements were found, especially after six months. Patients with fibromyalgic syndrome showed better response rates than those with orofacial neuropathic pain. Joint stiffness, masticatory fatigue, and sleep disturbances were also reduced. In conclusion, the local application of electromagnetic field appeared effective in treating fibromyalgic and neuropathic pain in the head and neck district, with broader improvements and no side effects.

## 1. Introduction

In the broad range of chronic pain disorders, neuropathic pain and fibromialgic syndrome (FS) represent common burdens affecting the worldwide population. Neuropathic pain, basically, describes a group of conditions with a wide range of causes and different pain distributions characterized by a lesion or disease affecting the somatosensory nervous system, both peripherally and centrally [1,2]. It affects between 6.9% and 10.0% of the general population [3], and only in the USA, 3.8 billion people suffer from it [4]. Orofacial neuropathic pain (ONP) develops from damage to the central nervous system (CNS) or trigeminal nerve [5]. It occurs with some features often reported by patients, such as electric shock, burning, cold, pricking, dye, and itching [6,7].

Similarly, FS represents an idiopathic and complex chronic syndrome, defined as long-lasting, widespread, and symmetric non-articular musculoskeletal pain with generalized tender points at specific anatomical sites [8,9]. FS is a common disorder affecting 1% to 5% of the general population [10], with a ratio male-female of 1:2 [11], occurring at any age [12]. Although pain is the dominant symptom, fatigue, non-refreshed sleep, mood disturbance, and cognitive impairment are common, worsening patients’ quality of life [13,14,15].

Therapeutic approaches should be directed toward an integrated treatment plan based on medications, patient education, physical activity, and cognitive-behavioral therapy [16,17,18]. Antidepressants, anticonvulsants, opioids, and topical products are mostly prescribed [19,20,21,22,23]. More invasive treatments, such as nerve blocks or surgeries, are recommended only in patients with pain refractory to conservative therapies, although their effectiveness is only partial [24]. Noninvasive brain stimulation techniques were demonstrated to be effective in the management of chronic pain [25,26,27]. Such techniques change the magnetic fields in order to increase or decrease neuronal activity, and they can be directed toward different targets [28]. It can be postulated that electromagnetic field (EMF) could be able to impact the morpho-functional processes of biological systems. Low and very low-frequency EMFs were demonstrated to induce functional changes in cells and tissues via influencing endogenous fields [29,30]. Nevertheless, there is a lack of evidence about methodologies and protocols in delivering EMF and about their effectiveness. The need to standardize therapeutic approaches based on EMF is therefore mandatory. In previous studies, several authors have shown positive effects of EMF on different targets, enhancing the production of collagen and cartilage glycosaminoglycans in fibroblasts [31] and chondrocytes [32], respectively, and improving osteoblasts’ osteogenic activity [33]. Furthermore, an effective modulation in the action of hormones and neurotransmitters on their specific receptors has been observed [34]. Similarly, a significant improvement in muscular performance was demonstrated [35], as well as a faster recovery of heart rate and serum lactate [36].

In the present scenario, the authors of the present study hypothesized the applicability of an emerging therapeutic protocol based on EMF therapy. A pilot study was designed in order to establish the background for subsequent more structured studies. A new medical device called Hologies-Orofacial Neuropathic Pain (HO-ONP) (Hologies Srl, 2159267, Cagliari, Italy) was used as adjuvant therapy to properly prescribed drugs in the treatment of pain in patients affected by FS or ONP.

## 2. Materials and Methods

The present study is an observational pilot study. The STROBE checklist guidelines have been followed. The initial sample was made by 28 individuals, sent to us by the Pain Center of the University of Cagliari. All the patients were diagnosed with FS or ONP. A total of 13 individuals were excluded after the application of inclusion and exclusion criteria. Inclusion and exclusion criteria are reported in Table 1. Patients to be included in the final sample needed to be (a) be at least 18 years old, (b) provide the signed informed consent, (c) suffering from FS diagnosed following the certificated criteria of the American College of Rheumatology [8] or chronic ONP, (d) have undergone proper diagnosis, (e) be under appropriate pharmacological treatment but without complete responsiveness; appropriate treatment needed to satisfy the latest evidence in terms of effectiveness for both FS and ONP [19,20,21,22,23]. Incomplete responsiveness to drugs meant no more than a partial recovery after 6 months of follow-up. Individuals of both genders were allowed to participate. By contrast, patients were excluded if they suffered from (a) psychiatric illnesses that preclude compliance to the therapy (memory disorders and psychiatric disorders not balanced), (b) temporomandibular joint disturbances according to the Axis I of Research and Diagnostic Criteria/Temporomandibular Disorders (RDC/TMD) [37], (c) other primary pains, (d) teeth-related odontalgia; patients addicted to psychotropic drugs and/or who were not under appropriate pharmacological treatment or with complete responsiveness to drugs were additionally excluded.

In the final sample, a total of 15 individuals were included, 10 with FS and 5 with ONP, all females, ranging from 28 years old up to 65. Two authors (MS and ES) examined and followed patients for the whole observational period, that was 12 months, at the Orofacial Pain Center of the University of Cagliari under the supervision of one independent coordinator (NZ). The kappa test was run to evaluate the level of agreement between these experts, and it was found to be 0.58, thus “moderate” [38]. Any disagreements were resolved through discussion between all the authors.

The aim of the study was to evaluate the effectiveness of HO-ONP in pain management in patients with ONP and FS as an adjunct therapy to prescribed drugs (see below). Additionally, the possible presence of any differences in the responsive rate between patients suffering from FS and those suffering from ONP were investigated.

The current study has been developed in agreement with the good clinical practice guidelines provided by the Declaration of Helsinki [39] and has been approved by the Medical Direction Board of the Surgical Sciences Department of the University of Cagliari (2020/16772).

The present pilot study has been developed according to the following 7 steps: (1) Medical examination: sites and origins of pain were evaluated; hyperalgesia and allodynia were searched by wiping a cotton roller and a pointed probe on the skin of both sides of the face and neck. The average level of pain felt was reported on the VAS scale. VAS scale is a 100 mm line anchored to a verbal description of pain from “no pain” to “the worst ever pain”. Patients are asked to mark that line approximately to indicate the level of pain [40]; patients were informed about the properties and risks of HO-ONP. Lastly, the signed informed consent was obtained.

Patients suffering from FS continued to follow the pharmacological protocol established previously by the Pain Center of the Hospital of the University of Cagliari: Duloxetine 30 mg/daily in the morning, eperisone hydrochloride 100 mg/once or twice per day, Clonazepam 5/7 drops before bedtime and magnesium 3 to 4.5 mg/kg per day. Patients with ONP continued instead to receive amitriptyline 12.5 to 25 mg/daily or pregabalin 150 to 300 mg/daily. (2) Application: the pulsed EMF was delivered through a small and flexible patch, 2 × 3 cm sized, that was comfortably wearable. The patch was applied by trained staff between C3 and C4 vertebra. Patients registered the pain level on the VAS scale at T0 (0 s). Patients were informed about the management of the patch and the need to change the position of the same sometimes to avoid the occurrence of tolerance. (3) At home, patients must have taken note of the pain level on the scale VAS at t_3_ (after 1 h), t_4_ (after 6 h), and t_5_ (before sleeping) for the first day only. Then, from the next day and for the entire duration of the follow-up, patients must have taken note at t_6_ (after awakening), at t_7_ (during the afternoon), and at t_5_ for each following day. (4) First check: the next week (T1), patients were recalled to the department to analyze potential rapid improvements in symptomatology and eventual unprogrammed side effects and to see if patients understood how to sign the VAS scale. (5) Second check: one month later (T2) to evaluate further improvements and eventual side effects. Then patients were recalled every month for the following year. (6) Third check: 6 months after the application (T3). (7) final check: after 12 months (T4) for the overall evaluation of the patch in the final detailed report.

Data were entered into a spreadsheet (MS Excel Office 365 MSO), and Gretl (gretl.sourceforge.net/, accessed on 10 January 2022) was the software used to perform the statistical analysis. ∆% was calculated through the analysis of median VAS scale values at each time point. The choice to carry out this type of test was taken by virtue of the low sample numerosity. The test does not require making assumptions about the distribution of the population.

The significance of changes in VAS score was evaluated by Friedman’s nonparametric two-way analysis of variance. Post-hoc analysis was performed by Wilcoxon signed-rank test with Bonferroni test. Statistical significance was set at *p* < 0.05, and analyses were performed using the statistical software Stata (release 16, StataCorp LLC, College Station, TX, USA).

## 3. Results

The current study analyzed the responsiveness of a total of 15 patients affected by FS or ONP to low-frequency EMF-based therapy. Median VAS scale values were registered at each check. Data were entered into a spreadsheet (MS Excel Office 365 MSO). The median values of the daily VAS scale were first calculated for each patient and then at T0, T1, T2, T3, and T4. Subsequently, a comparison was made, and the ∆% was calculated for each patient. ∆% resulted through the analysis and the comparison of each signed VAS scale of each patient, and it was calculated in order to verify whether the investigated treatment was effective. A global analysis was firstly performed, then patients were divided into two groups according to the disease they suffered from, namely FS or ONP, and ∆% was calculated for each group.

A decrease in pain levels in the overall sample was detected during the observational period between the initial stage and the following checks (Figure 1). Median VAS scale values, after the treatment was delivered, emerged notably lower than T0 (6.55), respectively 4.40 (T1), 4.03 (T2), 2.95 (T3), and 4.00 (T4).

The analysis provided via the Wilcoxon signed-rank test with Bonferroni revealed that overall changes in VAS score over time were significant (*p* < 0.001). Moreover, VAS score assessed at each time point significantly differ from VAS score measured at baseline (*p* = 0.002, *p* = 0.004, *p* = 0.010, and *p* = 0.010 at T1, T2, T3, and T4, respectively) (Figure 2).

Similar results appeared if considered separately the 10 patients affected by FS (Figure 3) to those 5 suffering from ONP (Figure 4). Most of the patients showed a decrease in pain intensity already a week after the application. Improvements were then confirmed in all the subsequent checks. Patients with FS showed more constant improvements compared to individuals suffering from ONP. Only 3 out of 10 patients with FS reported no pain improvements over the follow-up period, 2 of which referred even a worsening in symptoms. By contrast, half of the patients reported at least a 15% improvement in pain, four of which even a 25% of reduction. Among patients suffering from ONP, only one reported significant improvements in pain with up to 40% of reduction. Due to the low sample numerosity, no further tests analyzing data distribution could be assessed.

Patients with FS asserted furthermore benefits concerning several other symptoms. Reduced morning shoulder joint stiffness, partially but not completely, and diminished headaches, in both intensity and occurrence were reported by 8 out of 10 individuals. Additionally, masticatory fatigue was solved. Due to such improvements, the same patients reported having reduced physical therapy related to the neck and shoulders. Interestingly, significant self-reported enhancements in daily mood and in sleep were described. Reduced difficulties in falling asleep and lesser habitual nocturnal awakenings were the major clinical improvements referred by seven patients with both FS and ONP. In spite of such improvements, in patients with FS, the symptomatology related to all the other body regions was still present. Lastly, all the patients did not complain about the ergonomics and comfort of the therapeutic patch, and no side effects were reported.

## 4. Discussion

The purpose of the authors was to evaluate the effectiveness of the local administration of low-frequency EMF as an add-on therapy for chronic pain in patients with FS and ONP.

Low-frequency EMF therapy was demonstrated to be safe, noninvasive, and easy to use in the treatment of pain, with important effects in the control of different painful musculoskeletal disorders [41,42,43]. Specifically, the low-frequency electromagnetism-based techniques may lead to strong and repeated stimulations of synaptic junctions deleting the pathologic memory reflex pattern [42].

Foley-Nolan et al. [43], in a randomized trial, found that pulsed EMF was effective in reducing both pain and muscular stiffness after 6 weeks of treatment in patients with persistent neck pain. Similar outcomes were found when pulsed EMF was administered for the treatment of persistent rotator cuff tendinitis [42]. In a randomized, double-blind, sham-controlled clinical study, Sutbeyaz et al. [44] demonstrated that low-frequency EMF is able to improve mobility and decrease both pain and fatigue in patients with FS. Such evidence in FS was further confirmed by Sampson et al. [45].

In line with such evidence, we hypothesized the applicability of EMF as an adjuvant treatment in FS and ONP. In the current pilot study, the EMF is delivered through the communication among nanocrystals and semiconductors targeted with appropriate and desired signals. The specific signal is delivered through low electromagnetic frequencies and works in the oxy-hemoglobin wavelength by virtue of its ability to provide optimal noninvasive in vivo monitoring [46]. The signal has been built mirror-like to calcium ion with the aim to create a selected, localized, and controlled incoherence with the endogenous calcium itself. Such a mechanism may reduce calcium flow directly and subsequently the pro-nociceptive modifications induced in its channels both in the voltage-gated and in the ligand-gated. Indeed, the role of calcium channels in chronic pain has already been assessed and revised [47].

In the current study, the analysis of the results may conclude that EMF proved effective as an add-on therapy in the treatment of both FS and ONP. Important short-term and long-term benefits were reported by the majority of patients. A significant reduction in pain intensity was evident already after the therapy was administered. After 1 week (T1), a median 2 points-level improvement was assessed on the VAS scale.

Mild further improvements were reported after one month, while a significant additional decrease was observed after 6 months. During the final check, i.e., 12 months after the application, a 2.55 points-level improvement emerged through the analysis of the VAS scale. In line with clinical evidence, statistical significance emerged at each time point, *p* = 0.002, *p* = 0.004, *p* = 0.010, and *p* = 0.010 at T1, T2, T3, T4, respectively. It can be concluded that mild-to-moderate improvements were reported throughout the follow-up, with the major peak reached after six months. The progressive physical deterioration of the patch may be responsible for the improvements’ reduction at 12 months. Another hypothesis that should not be neglected is the possibility that a process of tolerance has been established. Future studies need to address this aspect.

Patients with FS appeared to respond better to EMF therapy than those with ONP. Assumed the very low numerosity of the ONP sample, only one out of five patients referred significant improvements in pain. Conversely, 40% of patients with FS reported an important reduction in pain. Furthermore, most treated patients with FS reported significant benefits regarding several other symptoms we initially did not consider. Rediscovered refreshed sleep, decreased headaches in intensity and occurrence, and reduced physical stiffness and functional limitations in movements were all reported. Patients additionally improved their daily mood, suggesting the primary impact on the quality of life of FS and ONP. Such results further supported the global improvements in pain, fatigue, and quality of life found by Sutbevaz et al. when pulsed EMF was delivered to patients with FS [44].

Additionally, a possible and direct influence of low-frequency EMF was suggested on the emotional component of pain [48]. Interestingly, but potentially expected, these effects have been felt only in treated parts, that is, above the shoulders, while the overall symptomatology remained unchanged in the other body regions. Due to such improvements, especially patients with FS reported a decrease in the physical therapy exercises usually performed to reduce cervical pain and an improvement in neck and shoulder movements.

Another outcome worthy of mention is the absence of any side effects reported since we asked patients to inform us continuously about each possible adverse event

The last parameter to consider regards the ergonomics of the device: all individuals undergone the study have not referred complaints about the comfort because the patch appeared to be easy to use, not awkward or bulky, and without any need for a power supply system such as other medical equipment working via electromagnetic waves.

The current study has several limitations due to the sample’s characteristics. Low numerosity is a major weakness. Furthermore, the limitation of providing only a localized effect gives rise to the need for further research. Indeed, the possibility to gain a systemic covering may lead to improvements in the quality of life of such patients and may potentially reduce the assumption of medications. The inclusion of only women in the final sample was expected because of epidemiological data related to both diseases.

## 5. Conclusions

In conclusion. albeit the low numerosity sample of our study and thus all the annexed limitations, we can support the benefits given by the local administration of low-frequency EMF in relieving head and neck pain in patients with both FS and ONP. However, patients suffering from FS appeared to better respond to the treatment. Additionally, a wider range of improvements concerning sleep, function, and mood also emerged in FS patients.

## Figures and Tables

**Figure 1 ijerph-19-04123-f001:**
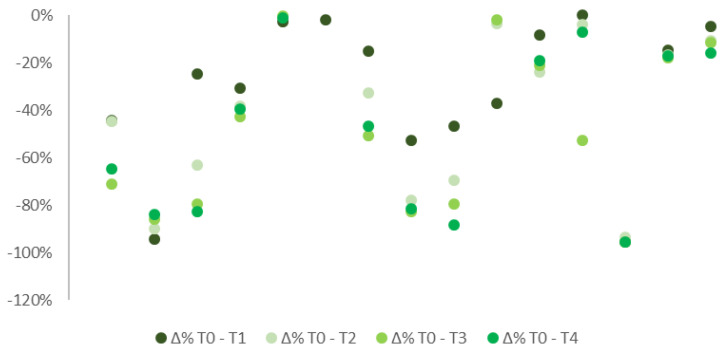
% difference in pain median levels at T0, T1, T2, T3, and T4 in all patients, with both FS and ONP.

**Figure 2 ijerph-19-04123-f002:**
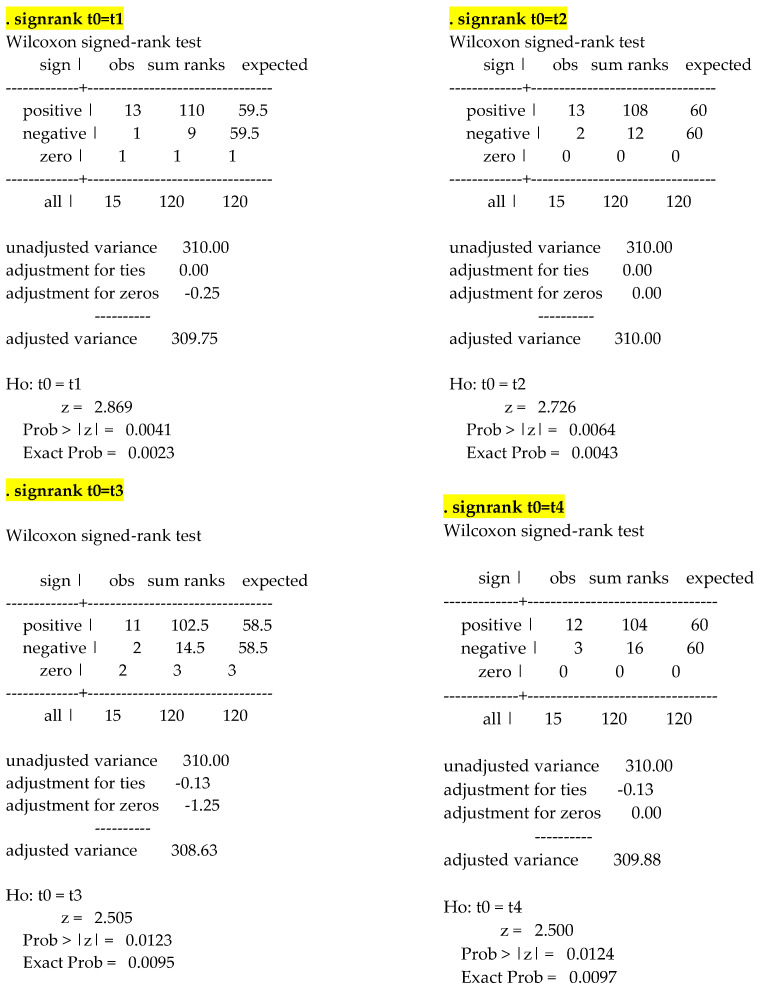
Friedman’s nonparametric two-way analysis of variance. Post-hoc analysis with Wilcoxon signed-rank test with Bonferroni.

**Figure 3 ijerph-19-04123-f003:**
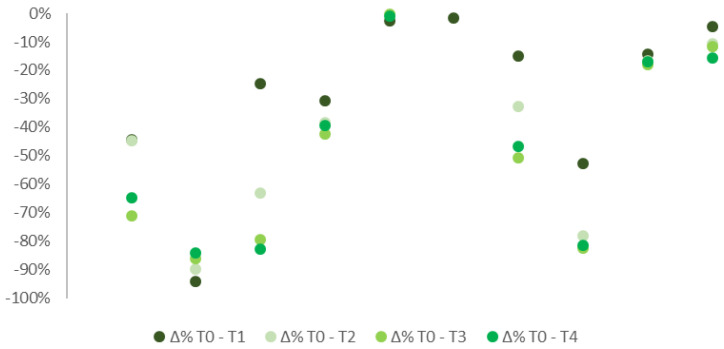
% differences in pain median levels at T0, T1, T2, T3, and T4 in patients with FS.

**Figure 4 ijerph-19-04123-f004:**
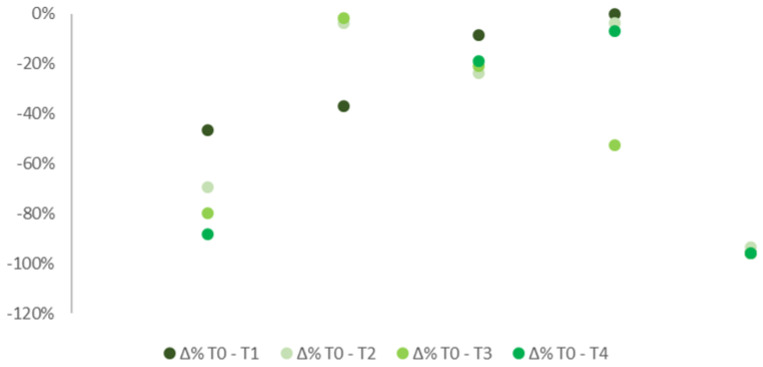
% differences in pain median levels at T0, T1, T2, T3, and T4 in patients with ONP.

**Table 1 ijerph-19-04123-t001:** Inclusion and exclusion criteria.

Inclusion Criteria	Exclusion Criteria
>18 years old; Both sexes; Provide the signed informed consent; Suffering from FS diagnosed following the certificated criteria of the American College of Rheumatology or chronic ONP; Diagnosed with FS or ONP by expert specialists; Be under appropriate pharmacological treatment, albeit without complete responsiveness.	Suffering from psychiatric illnesses that preclude compliance to the therapy (memory disorders and psychiatric disorders not balanced); Suffering from temporomandibular joint disturbances according to the Axis I of Research and Diagnostic Criteria/Temporomandibular Disorders (RDC/TMD); Suffering from other primary pains; Suffering from teeth-related odontalgia; Addiction to psychotropic drugs; Patients not under appropriate pharmacological treatment; Patients under appropriate pharmacological treatment with complete responsiveness to drugs.

## Data Availability

Data sharing not applicable due to patients’ privacy.

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
