# Peer review of "Local Administration of ElectroMagnetic Field as Add-On Therapy in the Treatment of Chronic Facial Pain: A Pilot Study"

_ijerph, 2022, doi:10.3390/ijerph19074123_

Round 1
Reviewer 1 Report
This preliminary study is highly commendable as it is a novel approach to topical application of electromagnetic fields for the treatment of chronic pain. Even though I disagree about the unification of fibromyalgia and orofacial neuropathic pain and classifying it as chronic facial pain, I think it is a very well-organized study in terms of study design and pain evaluation. I look forward to producing better results based on the results of this experiment.
Author Response
Dear reviewer,
Thank you for the appreciation and the suggestions. As you reported, the pilot study was necessary to better define the therapeutic protocol. We found extremely helpful your comment.
Reviewer 2 Report
Overall Impression
The authors conducted a pilot study to evaluate the effectiveness of low frequency EMF for chronic pain patients. Please find below some major points to consider.
Major Points
- To improve the reporting quality of your manuscript please follow the STROBE checklist reporting guidelines for observational studies. Link here: https://www.strobe-statement.org/checklists/
- Abstract, Lines 12-13: Please replace the background sentence with epidemiological data either prevalence or data from the global burden of disease studies. The second sentence is not appropriate since the primary outcome of interest should be pain and not depression (although it is important as secondary outcome).
- Introduction: The introduction is long, and it does not serve its purpose. Please clearly describe epidemiological data for the population of interest (use up to date references), how the intervention may it works and explain what the gap in the knowledge is and why this pilot study is needed. The introduction needs to be very specific and must be described with the latest references in maximum 1 page. Many sentences are not needed.
- Your research question is not clear. Evaluate the effectiveness on what outcome? I suspect is pain but you need to be very specific.
- Lines 78-79: The authors want to evaluate the effectiveness of HO-ONP as an adjuvant to another treatment. The other treatment needs to be described and included in the research question
- Please follow point by point the STROBE checklist for observational studies and submit the checklist in which page every item is located.
- Line 91: Did the authors indeed include gender? I think the authors are confused what constitutes gender because later in the results are stating that they included only females. Please correct.
- Line 100, Last Inclusion Criterion: How the authors evaluated this inclusion criterion? “be under appropriate pharmacological treatment albeit without a completely responsiveness.” What is appropriate? How the authors evaluated responsiveness on each pharmacological treatment for each patient that was screened for eligibility?
- Line 100, Third Exclusion Criterion: How the authors evaluated that the patients were suffering from other primary pains? What is other primary pains?
- Lines 103-107: The authors need to check this paragraph because it doesn’t make sense what they are stating. So, one author examined and followed-up the patients that was supervised from 2 independent coordinators. Then the authors calculate Kappa agreement for what? The agreement can be calculated between outcome assessors but it was 1 person. Then the coordinators did what exactly so the authors can calculate the kappa? Moderate means nothing if the authors do not present benchmark values.
- The intervention is not at all described. The intervention needs to be very specific with all details. What was the treatment duration? Frequency of the treatment? Sessions?
- How the authors ensured that the improvement was not due to pharmacological treatment and it was from the tested intervention?
- Statistical analysis is very poorly written and very poorly operationalized. Despite that the authors acknowledged the small sample size they have not present a statistical analysis plan at all.
- Your conclusion has only one aim and that is to answer the purpose of your study. Your conclusion is rather conversational than actual conclusion. Limitations and future clinical and research implications do not belong to the conclusion session.
Author Response
Dear reviewer,
thank you for the report and the suggestions, we found them extremely helpful to improve our manuscript. We provided the corrections suggested for every part of the study, as you can find in the revised manuscript. We revised the manuscript following the STROBE guidelines and its "statements". Methods and results were improved in order to be made more comprehensive. We explained some of our decisions/results through some comments you can find in the text. For ex. Line 91 regarding the gender, it was not a choice to include only females, by contrast we opened the study to both males and females; as both fibromyalgia and chronic pain are quite gender disorders, related to women, only some women met the requirements to be included in the final sample.
Abstract, Lines 12-13: suggested change has been made
Lines 78-79: we provided a deeper description of drugs prescribed to patients
Line 100 we hypotesized that the effect was related to the device and not to drugs( i.e. six months of follow up with only drugs without satisfactory improvements). We also better described the concept of "other primary pains"
We improved the statistic analysis provideng the Friedman and Wilcoxon tests with Bonferroni.
Lines 103-107 Kappa test was clarified and reported. It was a mistake during the writing process.
Introduction and conclusions were shorthened and modified to be more focused on the study's content. English levels was improved.
Reviewer 3 Report
Dear authors, it was with pleasure and interest that I could familiarize myself with the outcomes of your study. however, it needs some improvement before it can be published: Lines 50-52 please add 1 sentence as to why they represent a public health problem Line 62 – grammar Lines 67-68 what were the positive effects? Lines 78-81 add-on to what therapy? Lines 89-90 specify what expert specialists mean e) rephrase the sentence line 124 – magnesium – specify dose Conclusions 259-267 are not necessary Also, when it comes to "results" - it would add a lot if you could calculate whether there were any statistically significant differences in VAS scores at different points of time (*eg. for 2 groups eg T0 and T1 - Wilcoxon could be used, even though the sample size was small. This will make the study a little bit more comprehensible.
Author Response
Dear reviewer,
Thank you for the appreciation and the advices. We found extremely helpful your comments. We provided the suggested corrections, as you can find in the revised manuscript. The english level was ameliorated and a more comprehensive statistical analysis was provided using the Wilcoxon test.
Round 2
Reviewer 2 Report
I take the process of peer-review very seriously and I devote time and effort to provide a thorough and fair review and feedback.
Due to major limitation in the current study x I recommended “rejection” . Despite that, you sent me the paper for a second review without the authors to have addressed any of the major issues I raised.
I provided 14 points that need to be addressed by the authors, and the authors response cover only 4-5 of them.
Thus, I cannot provide any further recommendations until all my previous comments/points are addressed.
Author Response
Dear reviewer,
thank you for the report and the suggestions, we found them extremely helpful to improve our manuscript. We provided the corrections suggested for every part of the study, as you can find in the revised manuscript.
- To improve the reporting quality of your manuscript please follow the STROBE checklist reporting guidelines for observational studies. Link here: https://www.strobe-statement.org/checklists/
We revised the manuscript following the STROBE guidelines and its "statements". Statements were all followed and the manuscript was modified, where necessary, in order to accomplish STROBE statements. Some of them were further described in the below points.
- Abstract, Lines 12-13: Please replace the background sentence with epidemiological data either prevalence or data from the global burden of disease studies. The second sentence is not appropriate since the primary outcome of interest should be pain and not depression (although it is important as secondary outcome).
Abstract, Lines 12-13: suggested change has been made.
- Introduction: The introduction is long, and it does not serve its purpose. Please clearly describe epidemiological data for the population of interest (use up to date references), how the intervention may it works and explain what the gap in the knowledge is and why this pilot study is needed. The introduction needs to be very specific and must be described with the latest references in maximum 1 page. Many sentences are not needed.
Introduction was shortened and modified to be more focused on the study's content. Epidemiological data have been reported according to the most recent data. The impact of the intervention has been clarified through the presentation of the evidence regarding its use in other studies and describing the aim of the present study.
- Your research question is not clear. Evaluate the effectiveness on what outcome? I suspect is pain but you need to be very specific.
The aim of the study has been clarified and the research question has been described. They were introduced in the last part of the introduction and deeper clarified in the materials and methods.
- Lines 78-79: The authors want to evaluate the effectiveness of HO-ONP as an adjuvant to another treatment. The other treatment needs to be described and included in the research question
Lines 78-79: we provided a deeper description of drugs prescribed to patients (lines 141-146).
- Please follow point by point the STROBE checklist for observational studies and submit the checklist in which page every item is located.
The STROBE checklist was used as scaffold to support a better setting of the manuscript. Its statements are clarified in the text.
- Line 91: Did the authors indeed include gender? I think the authors are confused what constitutes gender because later in the results are stating that they included only females. Please correct.
Line 91 regarding the gender, it was not a choice to include only females, by contrast we opened the study to both males and females; as both fibromyalgia and chronic pain are quite gender disorders, related to women, only some women met the requirements to be included in the final sample.
- -9 Line 100, Last Inclusion Criterion: How the authors evaluated this inclusion criterion? “be under appropriate pharmacological treatment albeit without a completely responsiveness.” What is appropriate? How the authors evaluated responsiveness on each pharmacological treatment for each patient that was screened for eligibility?
Line 100, Third Exclusion Criterion: How the authors evaluated that the patients were suffering from other primary pains? What is other primary pains?
Line 100 we hypotesized that the effect was related to the device and not to drugs (i.e. six months of follow up with only drugs without satisfactory improvements). We also better described the concept of "other primary pains"
- Lines 103-107: The authors need to check this paragraph because it doesn’t make sense what they are stating. So, one author examined and followed-up the patients that was supervised from 2 independent coordinators. Then the authors calculate Kappa agreement for what? The agreement can be calculated between outcome assessors but it was 1 person. Then the coordinators did what exactly so the authors can calculate the kappa? Moderate means nothing if the authors do not present benchmark values.
Lines 103-107 Kappa test was clarified and reported. It was a mistake during the writing process.
- The intervention is not at all described. The intervention needs to be very specific with all details. What was the treatment duration? Frequency of the treatment? Sessions?
Methods and results were improved in order to be made more comprehensive the intervention. We explained some of our decisions/results through some comments you can find in the text.
- How the authors ensured that the improvement was not due to pharmacological treatment and it was from the tested intervention?
Line 106: only patients refractory to the described drugs for at least 6 months were included in the final sample in order to exclude the bias due to a potential effect of drugs.
- Statistical analysis is very poorly written and very poorly operationalized. Despite that the authors acknowledged the small sample size they have not present a statistical analysis plan at all.
We improved the statistical analysis providing the Friedman and Wilcoxon tests with Bonferroni. Significance of changes in VAS score were evaluated by the Friedman's nonparametric two-way analysis of variance. Post hoc analysis was performed by Wilcoxon signed-rank test with Bonferroni test. Statistical significance was set at p<0.05 and analyses were performed using the statistical software Stata (release 16, College Station, TX: StataCorp LLC).
- Your conclusion has only one aim and that is to answer the purpose of your study. Your conclusion is rather conversational than actual conclusion. Limitations and future clinical and research implications do not belong to the conclusion session.
Conclusions were deeply shortened and made more focus on the study’s results. Limitations were moved to Discussion.